# Less Descriptive yet Discriminative:
# Quantifying the Properties of Multimodal Referring Utterances via CLIP

**Ece Takmaz** and **Sandro Pezzelle** and **Raquel Fernández**

Institute for Logic, Language and Computation

University of Amsterdam

`{ece.takmaz|s.pezzelle|raquel.fernandez}@uva.nl`

## Abstract

In this work, we use a transformer-based pre-trained multimodal model, CLIP, to shed light on the mechanisms employed by human speakers when referring to visual entities. In particular, we use CLIP to quantify the degree of descriptiveness (how well an utterance describes an image in isolation) and discriminativeness (to what extent an utterance is effective in picking out a single image among similar images) of human referring utterances within multimodal dialogues. Overall, our results show that utterances become less descriptive over time while their discriminativeness remains unchanged. Through analysis, we propose that this trend could be due to participants relying on the previous mentions in the dialogue history, as well as being able to distill the most discriminative information from the visual context. In general, our study opens up the possibility of using this and similar models to quantify patterns in human data and shed light on the underlying cognitive mechanisms.

## 1 Introduction

During a conversation, speakers can refer to an entity (e.g., the girl in Fig. 1) multiple times within different contexts. This has been shown to lead to subsequent referring expressions that are usually shorter and that show lexical entrainment with previous mentions (Krauss and Weinheimer, 1967; Brennan and Clark, 1996). This trend has been confirmed in recent vision-and-language (V&L) datasets (Shore and Skantze, 2018; Haber et al., 2019; Hawkins et al., 2020): referring utterances become more compact (i.e., less descriptive), and yet participants are able to identify the intended referent (i.e., they remain pragmatically informative).

Several approaches (Mao et al., 2016; Cohn-Gordon et al., 2018; Schüz et al., 2021; Luo et al., 2018, i.a.) have tackled the generation of image captions from the perspective of pragmatic informativity; Coppock et al. (2020) have compared the

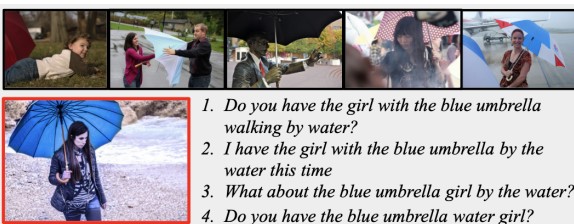

1. *Do you have the girl with the blue umbrella walking by water?*
2. *I have the girl with the blue umbrella by the water this time*
3. *What about the blue umbrella girl by the water?*
4. *Do you have the blue umbrella water girl?*

Figure 1: Referring utterance chain from PhotoBook (Haber et al., 2019). The chain has 4 ranks (4 references to the target image, in red outline). For simplicity, only the 5 distractor images from rank 1 are shown.

informativity of image captions and of referring expressions; and Haber et al. (2019); Hawkins et al. (2020) have explored how dialogue history contributes to discriminativeness. However, no work to date has investigated how these two dimensions, *descriptiveness* and *discriminativeness* or pragmatic informativity, interact in referring expressions uttered in dialogue.

In this work, we use a transformer-based pretrained multimodal model to study the interplay between descriptiveness and discriminativeness in human referring utterances produced in dialogue. Due to their unprecedented success in numerous tasks, pretrained V&L models—such as LXMERT (Tan and Bansal, 2019), VisualBERT (Li et al., 2019), UNITER (Chen et al., 2020) and ALIGN (Jia et al., 2021)—have recently attracted a lot of interest aimed at understanding the properties and potential of their learned representations as well as the effect their architectures and training setups have (Bugliarello et al., 2021). These include probing such models in a zero-shot manner, i.e., without any specific fine-tuning (Hendricks and Nematzadeh, 2021; Parcalabescu et al., 2021); quantifying the roles of each modality (Frank et al., 2021); inspecting attention patterns (Cao et al., 2020); and evaluating their learned multimodal representations against human judgments (Pezzelle et al., 2021).

We focus on one model: Contrastive Language-

Image Pre-training (CLIP, Radford et al., 2021), which learns via contrasting images and texts that can be aligned or unaligned with each other. This contrastive objective makes CLIP particularly suitable for modelling referential tasks that inherently include such comparisons. Here, we use CLIP to gain insight into the strategies used by humans in sequential reference settings, finding that although the descriptiveness of referring utterances decreases significantly, the utterances remain discriminative over the course of multimodal dialogue. The code to reproduce our results is available at `https://github.com/ecekt/clip-desc-disc`.

## 2 Data

We focus on PhotoBook (PB; Haber et al., 2019), a dataset of multimodal task-oriented dialogues where players aim to pick the images they have in common without seeing each other's visual contexts (which consist of 6 images coming from the same domain). The game is played over several rounds in which the previously seen images reappear in different visual contexts, giving the players an opportunity to refer to such images again. As a result, *chains* of utterances referring to a single image are formed over the rounds as the players build common ground. See Fig. 1 for a simplified representation of a chain.[1] In total, PB consists of 2,500 games, 165K utterances, and 360 unique images from COCO (Lin et al., 2014).

All our experiments are conducted on a subset of 50 PB games with manually annotated referring utterances, which contains 364 referential chains about 205 unique target images. We refer to this subset as PB-GOLD.[2] Although a dataset of automatically-extracted chains using all PB data is also available (Takmaz et al., 2020), as reported by the authors these chains may contain errors. We therefore opt for using the smaller but higher-quality PB-GOLD subset since we are interested in analysing human strategies. Given that we use a pretrained model without fine-tuning, experimenting with large amounts of data is not a requisite.

PB-GOLD's chains contain 1,078 utterances, i.e., 2.96 utterances per chain on average (min 1, max 4). We henceforth use the term 'rank' to refer to the position of an utterance in a chain. The average

token length of utterances is 13.34, 11.03, 9.23, and 7.82, respectively, for ranks 1, 2, 3, and 4.[3] This decreasing trend, which is statistically significant at $p < 0.01$ with respect to independent samples t-tests between the ranks, is in line with the trend observed in the whole dataset (Haber et al., 2019). PB-GOLD's vocabulary consists of 926 tokens.

## 3 Model

We use CLIP (Radford et al., 2021), a model pretrained on a dataset of 400 million image-text pairs collected from the internet using a contrastive objective to learn strong transferable vision representations with natural language supervision.[4] In particular, we employ the ViT-B/32 version of CLIP, which utilizes separate transformers to encode vision and language (Vaswani et al., 2017; Dosovitskiy et al., 2021; Radford et al., 2019, 2021).

As the model learns to align images and texts, this enables zero-shot transfer to various V&L tasks such as image-text retrieval and image classification and even certain non-traditional tasks in a simple and efficient manner (Radford et al., 2019; Agarwal et al., 2021; Shen et al., 2021; Cafagna et al., 2021; Hessel et al., 2021). This makes it an intriguing tool to investigate the properties of visually grounded referring utterances. In this work, we freeze CLIP's weights and do not fine-tune the model or perform prompt engineering, since we aim to exploit the model's pretrained knowledge for the analysis of human referring strategies.

## 4 Descriptiveness

In our first experiment, we investigate the degree of descriptiveness exhibited by referring utterances in the PhotoBook game, i.e., the amount of information they provide about the image out of context. We consider each target image and corresponding referential utterance at a given rank *in isolation*, i.e., without taking into account the other competing images nor the dialogue history. We quantify descriptiveness as the alignment between an utterance and its image referent using `CLIPScore` (Hessel et al., 2021), assuming that a more descriptive utterance will attain a higher score. For all the target image-utterance pairs in the chains of PB-GOLD, we use CLIP to obtain a vector $t$ representing the utterance and a

---

[1] Only 1 player's perspective for 1 context is represented.
[2] We use the gold set of the utterance-based chains v2 available at `https://dmg-photobook.github.io/`.

[3] We use TweetTokenizer: `https://www.nltk.org/api/nltk.tokenize.html`
[4] `https://github.com/openai/CLIP`

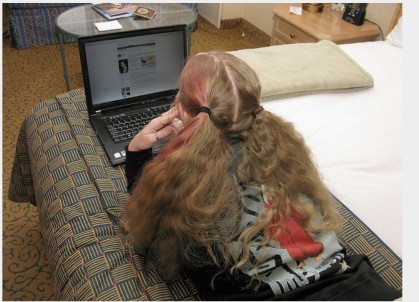

1. *girl lying on a bed surfing the internet on a laptop computer*
2. *a girl sleeping on her belly on top of a bed looking at a laptop.*
3. *woman laying on her stomach on a bed in front of a laptop.*
4. *a girl with long brown hair with streaks of red lays on a bed and looks at an open laptop computer.*
5. *a young girl laying on a bed using her laptop.*

Figure 2: Set of captions from COCO (Lin et al., 2014), the order of captions is arbitrary.

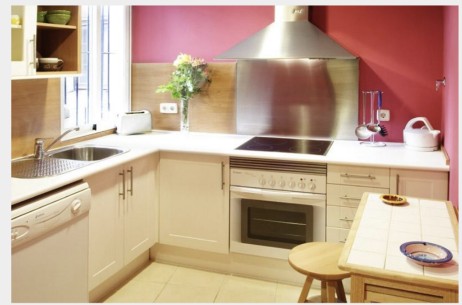

1. *This is a kitchen with white cupboards and worktop.*
2. *There is a red wall painted behind the cooker section.*
3. *There is a wooden table to the right with pale floor tiles on the floor.*
4. *There is a sink to the left with a window near the sink.*
5. *There is a bunch of flowers beside the window.*

Figure 3: Sequential description from Image Description Sequences (Ilinykh et al., 2019).

vector $v$ representing the image. CLIPScore is then computed as the scaled cosine similarity between these two vectors, with range $[0, 2.5]$:[5]

CLIPScore$(t, v) = 2.5 * max(cos(t, v), 0)$. We compute the average CLIPScore per rank over the whole PB-GOLD dataset.

**Results.** We find that earlier utterances are better aligned with the target image features and that there is a monotonically decreasing trend over the 4 ranks (Fig. 4, blue bars). The differences between all pairs of ranks are statistically significant (according to independent samples t-tests, $p < 0.01$), except for the comparison between the last 2 ranks ($p > 0.05$). Since earlier referring utterances tend to be longer (see Sec. 2), we check to what extent length may be a confounding factor. We find that there is only a weak correlation between token length and CLIPScore (Spearman's $\rho = 0.29, p < 0.001$).

We compare these results on PhotoBook with text-to-image alignment computed with the same method on two other datasets: (1) COCO (Lin et al., 2014),[6] which includes 5 captions per image provided independently by different annotators as shown in Fig. 2; here we do not expect to find significant differences in the level of descriptiveness across the captions, and (2) Image Description Sequences (IDS, Ilinykh et al., 2019)[7] where one participant describes an image incrementally as shown in Fig. 3, by progressively adding sentences with further details; here we do expect a similar

pattern to PhotoBook, albeit for different reasons (because participants add less salient information; Ilinykh et al., 2019).

Fig. 4 shows that these expectations are confirmed. According to CLIP, COCO captions (green bars) are more descriptive than IDS descriptions and PB referring utterances, and are equally aligned with the image across 'ranks' (the order is arbitrary in this case). In contrast, IDS incremental descriptions (yellow bars) are intrinsically ordered and show a significant decreasing trend similar to PB.

## 5 Discriminativeness

In order for a listener to select the target image among distractor images, a referring utterance should be discriminative in its visual context. Our results in the previous section show that descriptiveness decreases over time—what is the trend regarding discriminativeness? To address this question, in our second experiment we use CLIP from the perspective of reference resolution.

We focus on local text-to-image alignment, initially ignoring the previous dialogue history. To this end, we feed CLIP a single referring utterance together with the visual context of the speaker who produced that utterance. CLIP yields softmax probabilities for each image contrasted with the single text. As a metric, we use accuracy: 1 if the target image gets the highest probability; 0 otherwise.

**Results.** The overall accuracy is 80.15%, which is well above the random baseline of 16.67%. In Fig. 5, we break down the results per rank (blue bars). A $4 \times 2$ chi-square test (4 ranks vs. correct/incorrect) did not yield significant differences

---

[5]The scaled factor was introduced by Hessel et al. (2021) to account for the relatively low observed cosine values.

[6]We use the set of COCO images in PB-GOLD ($N$=205).

[7]The images are from ADE20k corpus (Zhou et al., 2017)

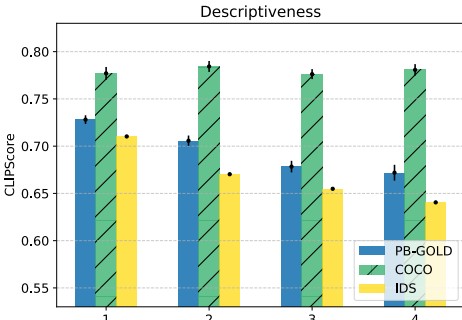

Figure 4: Descriptiveness (`CLIPScore`) for PB-GOLD, COCO and IDS. We only plot the first 4 'ranks' (x-axis) for COCO and IDS for comparability with PB-GOLD. The error bars illustrate the standard error.

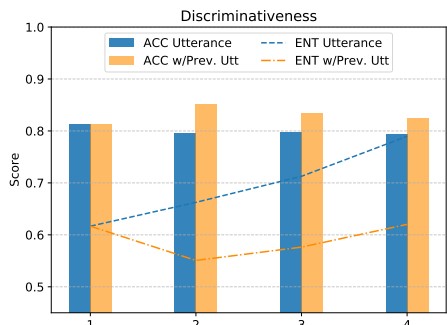

Figure 5: Discriminativeness (reference resolution accuracy, ACC) per rank with PB-GOLD utterances (Utterance) and utterances with history (w/Prev. Utt), along with their respective entropies (ENT).

in accuracy between the ranks, $p > 0.05$. Thus, although descriptiveness decreases over time, discriminativeness is not significantly affected. An analysis of the entropy of the softmax distributions reveals that entropy increases monotonically over the ranks (this difference is statistically significant according to an independent samples t-test between ranks 1 and 4; $H_1 = 0.62$, $H_4 = 0.79$, $p < 0.01$). That is, the model is more uncertain when trying to resolve less descriptive utterances. There is indeed a negative correlation between entropy and `CLIPScore` computed between the target image and the corresponding utterance (Spearman's $\rho = -0.5, p < 0.001$).

## 6 Analysis

How do participants manage to maintain discriminativeness while decreasing descriptiveness? Do they rely on the previous mentions present in the dialogue history? Do they refine their referring strategy by distilling the most discriminative information in a given context?

### 6.1 Dialogue history

The results of our experiment in the previous section show that the utterances in isolation are effective at referring; yet, uncertainty increases when the less descriptive utterances are considered out of context. To reduce such uncertainty, participants may rely on the dialogue history (Brennan and Clark, 1996; Shore and Skantze, 2018; Takmaz et al., 2020). We consider a scenario where participants keep in memory the previous mention when processing the current referring utterance. We model this scenario by prepending the previous referring utterance in the chain to the current utterance and feeding this into the reference reso-

lution model described in Section 5. As shown in Fig. 5, the resulting discriminativeness is similar to the one obtained earlier (the differences are not significant; chi-square test, $p < 0.05$) and, as before, remains stable across ranks (chi-square test, $p > 0.05$). However, taking into account the previous mentions leads to a significant reduction of the entropy in general: e.g., at the last rank $H_4 = 0.79$ vs. $H'_4 = 0.62$ (t-test, $p < 0.05$). This suggests that relying on the dialogue history allows speakers to use less descriptive utterances by reducing discriminative uncertainty.

### 6.2 Most discriminative information

Besides exploiting the dialogue history, participants may refine their referring strategy by distilling the most discriminative information in a given context. To gain insight into this hypothesis, we explore what is discriminative in the images: we compute the discriminative features $v_d$ of a target image by taking the average of the visual representations of distractor images to obtain the mean context vector and then subtracting this vector from the visual representation of the target image. We encode all 926 words in the vocabulary of PB-GOLD using CLIP, and retrieve the top-10 words whose representations are the closest to $v_d$ in terms of cosine similarity (amounting to 1% of the vocabulary). We take these words to convey the most discriminative properties of an image in context. We analyse whether at least one of these retrieved words is mentioned exactly in the referring utterance, finding that this is indeed the case for a remarkable 60% of utterances.[8] As an illustration, for the example in Fig. 1, the words *walking* (mentioned at rank 1)

---

[8]Randomly sampling 10 words from the vocabulary for each utterance yields 11% (average of 5 random runs).

and *blue* (used at ranks 1, 2, 3, 4) are among the top-10 most discriminative words, while the word *water* (mentioned at ranks 1, 2, 3, 4) is close to the word *beach*, which is also retrieved as one of most discriminative words in this case.

The most discriminative words are likely to be reused in later utterances, even though the visual context changes from rank to rank. For instance, the most discriminative words mentioned at rank 1 constitute 60% of the discriminative words at rank 2, indicating that entrainment is likely for words that have high utility across contexts. We also find a significant increase in the proportion of discriminative content words to all the content words per utterance (only between ranks 1 and 4, 14% vs. 19%, $p < 0.01$).

## 7 Conclusion

We used a pre-trained multimodal model claimed to be a reference-free caption evaluator, CLIP (Radford et al., 2021), to quantify descriptiveness and discriminativeness of human referring utterances within multimodal dialogues. We showed that (i) later utterances in a dialogue become less descriptive in isolation while (ii) remaining similarly discriminative against a visual context.

We found that the addition of dialogue history helps decrease and control the entropy of resolution accuracy even when the speakers produce less descriptive referring utterances. In addition, we found that the proportion of discriminative words increases over the ranks. These suggest that participants playing the PhotoBook game (Haber et al., 2019) show a tendency towards distilling discriminative words and utilize the dialogue history to keep task performance stable over the dialogue. This outcome resonates with the findings by Giulianelli et al. (2021) who observe that PhotoBook dialogue participants tend to limit fluctuations in the amount of information transmitted within reference chains, in line with uniform information density principles (e.g., Genzel and Charniak, 2002; Jaeger and Levy, 2007).

Interestingly, future work could explore novel ways of incorporating the CLIP model or its representations into a reference resolution or generation model embedding dialogue history and visual context to obtain human-like outcomes.

## Acknowledgments

We would like to thank Mario Giulianelli and Arabella Sinclair for their valuable comments on a draft of this paper. This project has received funding from the European Research Council (ERC) under the European Union's Horizon 2020 research and innovation program (grant agreement No. 819455).

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
