# OpenReview forum: "Less Descriptive yet Discriminative: Quantifying the Properties of Multimodal Referring Utterances via CLIP"
_aclweb.org/ACL/2022/Workshop/CMCL — CMCL 2022_

### Official Review · Reviewer_YG2K · 2022-03-21
**Nice work. Applies state-of-the-art tools in a simple and effective manner in order to measure human language usage in a conversation.**

**Rating:** 7
**Confidence:** 4

**Review:**

The paper uses the CLIP language-vision model in order to measure the descriptiveness (how well an utterance describes an image in isolation) and discriminativeness (to what extent an utterance is effective in picking out a single image among similar images) of human language as it develops in a conversation.  The focus is on utterances that refer to visual entities. The paper presents interesting conclusions, reasonable explanations and it paves the way for future research with CLIP and similar tools.

---

### Official Review · Reviewer_yWif · 2022-03-25
**An interesting study on multimodal referring utterances**

**Rating:** 7
**Confidence:** 4

**Review:**

This paper presents experiments on quantifying the descriptiveness and discriminativeness of referring utterances in dialogues, by using a pre-trained multimodal modal. The study is well-presented and has shown interesting results. I only have one minor question about the COCO dataset. It says that the COCO set used in this study was a part of the PB-GOLD, but while PB-GOLD's utterances were related (section 2), later it says COCO dataset's captions were independently produced (page 3). Maybe some clarification would help us understand the nature of PB-GOLD dataset.

---

### Official Review · Reviewer_6fTE · 2022-03-27
**Investigating multimodal referring items using transformers**

**Rating:** 6
**Confidence:** 3

**Review:**

In this paper, the authors use CLIP (a multimodal vision + language neural model) to quantify the descriptiveness and discriminative of human dialogues when describing visual inputs.

The work is interesting, and overall well presented. The only part I would like spelled out better is the ability to really get cognitive insights into the possible strategies adopted by humans. So far, the link is really not there and unfortunately, undermines the conclusions we can draw from this study.

---

### Decision · Program_Chairs · 2022-03-29

Accept